# A shape-shifting redox foldase contributes to *Proteus mirabilis* copper resistance

Emily J. Furlong[1], Alvin W. Lo[2,3], Fabian Kurth[1,†], Lakshmanane Premkumar[1,2,†], Makrina Totsika[2,3,†], Maud E.S. Achard[2,3,†], Maria A. Halili[1], Begoña Heras[4], Andrew E. Whitten[1,†], Hassanul G. Choudhury[1,†], Mark A. Schembri[2,3] & Jennifer L. Martin[1,5]

Copper resistance is a key virulence trait of the uropathogen *Proteus mirabilis*. Here we show that *P. mirabilis* ScsC (PmScsC) contributes to this defence mechanism by enabling swarming in the presence of copper. We also demonstrate that PmScsC is a thioredoxin-like disulfide isomerase but, unlike other characterized proteins in this family, it is trimeric. PmScsC trimerization and its active site cysteine are required for wild-type swarming activity in the presence of copper. Moreover, PmScsC exhibits unprecedented motion as a consequence of a shape-shifting motif linking the catalytic and trimerization domains. The linker accesses strand, loop and helical conformations enabling the sampling of an enormous folding landscape by the catalytic domains. Mutation of the shape-shifting motif abolishes disulfide isomerase activity, as does removal of the trimerization domain, showing that both features are essential to foldase function. More broadly, the shape-shifter peptide has the potential for 'plug and play' application in protein engineering.

[1] Institute for Molecular Bioscience, University of Queensland, St Lucia, Queensland 4072, Australia. [2] School of Chemistry and Molecular Biosciences, University of Queensland, St Lucia, Queensland 4072, Australia. [3] Australian Infectious Diseases Research Centre, University of Queensland, St Lucia, Queensland 4072, Australia. [4] La Trobe Institute for Molecular Science, La Trobe University, Bundoora, Victoria 3068, Australia. [5] Griffith Institute for Drug Discovery, Griffith University, Nathan, Queensland 4111, Australia. † Present addresses: Bristol-Myers Squibb, Arnulfstraße 29, 80636 Munich, Germany (F.K.); Department of Microbiology and Immunology, School of Medicine, University of North Carolina, Chapel Hill, North Carolina 27514, USA (L.P.); Institute of Health and Biomedical Innovation, School of Biomedical Sciences, Queensland University of Technology, Kelvin Grove, Queensland 4059, Australia (M.T.); School of Human Movement and Nutrition Sciences, University of Queensland, St Lucia, Queensland 4072, Australia (M.E.S.A.); Australian Centre for Neutron Scattering, Australian Nuclear Science and Technology Organization, Lucas Heights, New South Wales 2234, Australia (A.E.W.); Cello Health Consulting, Farnham Surrey GU9 7DN, UK (H.G.C.). Correspondence and requests for materials should be addressed to A.E.W. (email: awh@ansto.gov.au) or to M.A.S. (email: m.schembri@uq.edu.au) or to J.L.M. (email: jlm@griffith.edu.au).

The *Escherichia coli* periplasmic Dsb protein family is the best characterized bacterial oxidative folding system. *E coli* DsbA is a monomeric thioredoxin-fold oxidase[1] that introduces disulfide bonds into protein substrates[2], whereas *E. coli* DsbC is a dimeric V-shaped thioredoxin-fold protein disulfide isomerase[3] that proof-reads and shuffles incorrect disulfide bonds[4]. *E. coli* DsbG is also a dimeric V-shaped thioredoxin-fold protein disulfide isomerase and cysteine reducing system[5] that protects cysteines of periplasmic proteins from inappropriate oxidation[6].

Here, we characterize a putative DsbA-like protein from *P. mirabilis* and show it plays a key role in virulence by enabling swarming during copper stress. We find that it is not DsbA-like, as it does not catalyse disulfide bond formation. Rather, it is a disulfide isomerase that shuffles incorrect disulfide bonds in proteins. Further, we show that this thioredoxin fold protein is unlike any other characterized to date—it is trimeric—and we demonstrate that its function depends on a shape-shifting motif that could potentially be used as a 'plug and play' peptide module.

## Results

*Proteus mirabilis* UniProt C2LPE2 is a predicted thioredoxin-fold protein, and the purified protein exhibits redox properties characteristic of the thioredoxin fold family. It is highly oxidizing ($-108$ mV), has an acidic active site cysteine ($pK_a$ 3.1; compared to a typical cysteine thiol pKa of $\sim 8$–9) and the oxidized disulfide form of the active site destabilizes the protein compared with the reduced form (Supplementary Fig. 1a–c).

The encoded protein plays an important role in *P. mirabilis* virulence. Copper intoxication is a component of nutritional immunity having a number of detrimental effects on bacterial cells. Copper can react with host-generated hydrogen peroxide to generate free radicals, which damage biological molecules[7]; cycling between Cu(I) and Cu(II) can lead to detrimental redox reactions in bacteria; and copper can bind to protein thiols in place of other metal co-factors[8,9]. Overcoming these antibacterial effects of copper is a key bacterial defence mechanism. The gene encoding *P. mirabilis* C2LPE2 is located within a cluster of four predicted **s**uppressor of **c**opper **s**ensitivity (Scs) genes and we proposed this the encoded protein (hereafter called PmScsC) would contribute to bacterial copper resistance. Indeed, inactivation of *scsC* in two independent *P. mirabilis* clinical isolates significantly inhibited swarming motility (a key virulence trait of *P. mirabilis*) in the presence of copper (Fig. 1b, Supplementary Figs 2–4). This phenotype could be complemented by introduction of a plasmid encoding wild-type PmScsC, but not a plasmid encoding an inactive PmScsC (Fig. 1a,b).

Although it is annotated DsbA-like, PmScsC is a powerful protein disulfide isomerase (Fig. 1c) that is able to refold and reactivate the scrambled disulfide form of the model substrate RNase A just as rapidly as the archetypal disulfide isomerase *E. coli* DsbC. It has negligible DsbA-like dithiol oxidase activity (Supplementary Fig. 1d).

The best-characterized protein disulfide isomerases—EcDsbC and eukaryotic protein disulfide isomerase (PDI)—each have two thioredoxin-fold catalytic domains. DsbC is a dimer, and PDI is a modular 4-domain protein with two catalytically active thior-edoxin domains. The presence of two thioredoxin catalytic domains is thought to contribute to highly efficient disulfide shuffling activity[10,11]. The amino acid sequence of PmScsC encodes a single thioredoxin fold, and we therefore expected that its disulfide isomerase activity would be a consequence of dimerization to generate the necessary two catalytic domains. We also expected that the protein would adopt the same dimeric

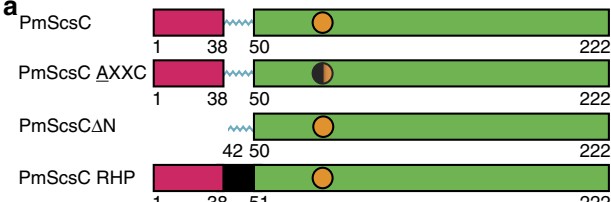

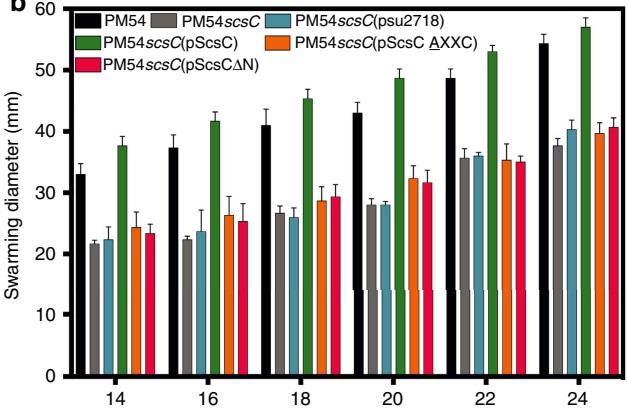

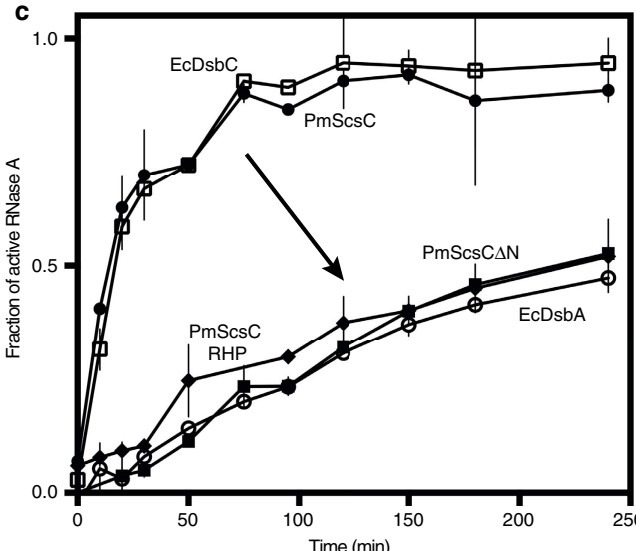

**Figure 1 | PmScsC function.** (**a**) Linear representation of the domain organization of PmScsC mutants. The trimerization domain is coloured magenta, the catalytic domain is green, with the CXXC motif represented as an orange circle, and the linker region is shown in cyan. The rigid helical linker is shown in black. (**b**) Swarming motility of wild type *P. mirabilis* strain PM54 (black), PmScsC deletion mutant PM54*scsC* (grey) and PM54*scsC* containing control and complementation plasmids: PM54*scsC*(pSU2718) (vector control; cyan), PM54*scsC*(pScsC) (wild-type; green), PM54*scsC* (pScsC ΔXXC) (active site mutant; orange) and PM54*scsC*(pScsCΔN) (N-terminal trimerization domain deletion; magenta) in the presence of 1.5 mM CuSO₄ on LB agar plates (significant difference, $P < 0.0001$ for slope calculated by F-test). Data are shown as the mean ± s.d., of a single experiment performed in triplicate; all data is representative of three independent experiments. (**c**) Disulfide isomerase activity of PmScsC (hollow squares) is similar to that of EcDsbC (filled circles). Monomeric PmScsΔN (filled squares) and the PmScsC mutant engineered to have a rigid helical peptide (RHP) in place of the flexible peptide (filled diamonds) show negligible activity (equivalent to oxidase EcDsbA (hollow circles)). Data are shown as the mean ± s.d., of two replicate experiments.

architecture as V-shaped EcDsbC. However, we were wrong on both counts.

Unexpectedly, evidence from chemical cross-linking (Supplementary Fig. 5), small angle X-ray solution scattering (Fig. 2a,b) and multi-angle light scattering (Fig. 2c) all indicated that PmScsC is trimeric. We confirmed that PmScsC is a trimer by determining three independent crystal structures. The three crystal structures—which we refer to as compact, transitional and extended (Fig. 3a–c, Table 1)—reveal an extraordinary range of motion in this protein. Importantly, the compact and transitional structures have multiple protomers in the asymmetric unit that adopt the same overall structures in each case (8 compact trimers, RMSD ∼1.5 Å for 645 Cα; two transitional trimers, RMSD 0.5 Å for 653 Cα) so that crystal symmetry does not appear to impact on the observed conformation. Moreover, the compact and transitional conformations were determined from crystals grown under similar conditions (2.85 M sodium malonate pH 5.8, 20 °C with either 0.1 M cobalt or copper added). Nevertheless, comparison of the eight compact and the two transitional trimers in these two crystal structures, reveals an unprecedented level of conformational re-arrangement (RMSDs > 20 Å). The third structure was determined from crystals grown in the presence of 2 mM copper chloride (32% Jeffamine M-600, 0.1 M HEPES pH 8, 20 °C) and reveals a fully extended trimer with one protomer in the asymmetric unit.

PmScsC also exhibits dynamic conformational flexibility in solution. No single PmScsC crystal structure is consistent with the experimentally determined SAXS scattering curve. However, an ensemble of structures fits the experimental data closely (Fig. 2a,b) supporting the notion that the trimer is highly dynamic in solution. Moreover, the solution scattering curve is unchanged by modification of the pH (range 6.0–8.0) or ionic strength (150–1,500 mM NaCl) (Supplementary Fig. 6) indicating the dynamic motion is an inherent property of the protein.

We can pinpoint the region responsible for the extraordinary range of motion in this protein, to an 11-residue peptide linking the trimerization and catalytic domains (residues 39-KADEQ-QAQFRQ-49) (Fig. 3). The 31 crystallographically characterized PmScsC protomers have the same thioredoxin-fold catalytic domains (Fig. 4a,b) (RMSD 0.2–1.2 Å for 129 Cα). Similarly, all 31 PmScsC protomers share the same trimeric right-handed coiled coil stalk formed from the N-terminal residues (Fig. 3a–c). At the secondary structure level, it is only the 11-residue linker peptide that changes significantly across the crystal structures (Fig. 4c, Supplementary Movie 1).

The linker is a shape-shifting peptide that can adopt helical, strand or loop conformations. In the 24 protomers present in the compact PmScsC crystal structure, the linker forms a loop that positions the three catalytic domains close to the trimerization stalk (Figs 3a and 4c). In the extended crystal structure the linker is helical, rotating and translating the catalytic domain away from the trimerization domain relative to the compact structure (Figs 3c and 4c). In the transitional PmScsC crystal structure, each of the two trimers incorporates protomers in three conformations: compact, extended-like and intermediate (Fig. 3b). The two compact protomer conformations of the transitional structure are similar to those in the compact crystal structure (RMSD 0.9–1.8 Å for 188 Cα). The two extended-like protomer conformations of the transitional structure are similar to the protomer in the extended crystal structure though they bend at different points in the linker helix, giving rise to somewhat different catalytic domain placements (Fig. 4c). The two intermediate conformations in the transitional crystal structure have a linker that forms a short β-strand, and positions the catalytic domain directly above the trimerization stalk (Fig. 4c).

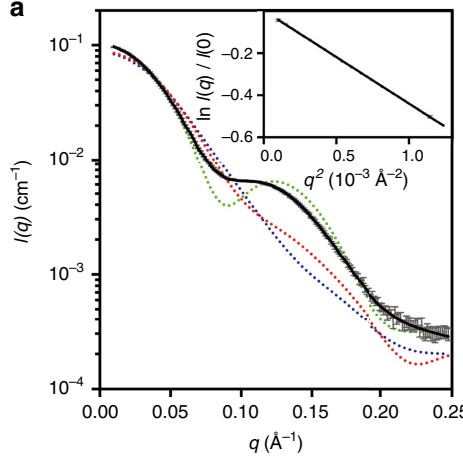

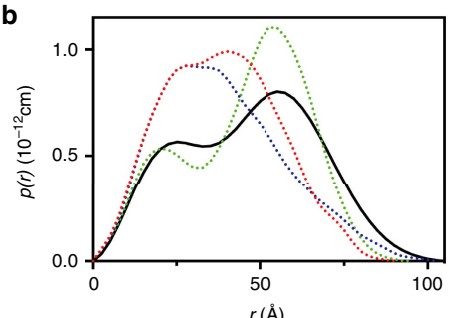

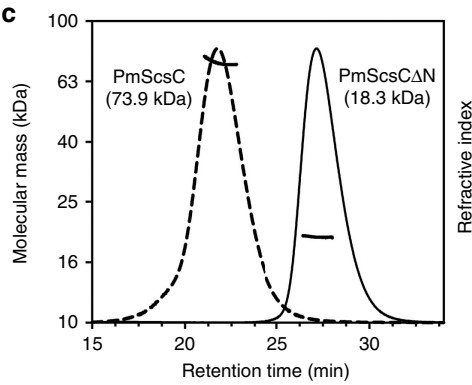

**Figure 2 | SAXS and MALLS of PmScsC.** (**a**) Small-angle X-ray scattering data collected from wild type PmScsC (grey) and the calculated scattering profile of the ensemble model overlayed in black (SASBDB: SASDB94). The predicted scattering profile of each of the crystal structures is also shown (dashed lines: PDB: 4XVW compact, red; PDB: 5IDR transitional, blue; PDB: 5ID4 extended, green). The agreement between the experimental data and the ensemble model is excellent, yielding $\chi^2 = 1.0$ (compared to $\chi^2 = 863.9$ (compact); $\chi^2 = 1222$ (transitional); $\chi^2 = 348.2$ (extended)). The Guinier region (inset) of the scattering data is linear, consistent with a monodisperse solution. (**b**) Pair distance distribution function derived from the scattering data, showing the maximum dimension of the particles in solution is 105 Å. Also shown is the calculated $p(r)$ for each of the crystal structures (dashed lines: compact, red; transitional, blue; extended, green), showing a maximum dimension of 90, 105 and 100 Å, respectively. The $p(r)$ generated from the extended structure (green) is most similar to the experimentally derived $p(r)$, while the other $p(r)$ curves are markedly different. (**c**) MALLS profile of PmScsC and PmScsCΔN. PmScsC eluted faster (21.5 min) than PmScsCΔN (27.2 min): experimentally determined molecular masses are 74 ± 0.3 kDa for PmScsC (theoretical trimer mass 74.3 kDa) and 18.3 ± 0.3 kDa for PmScsCΔN (theoretical monomer mass 18.1 kDa).

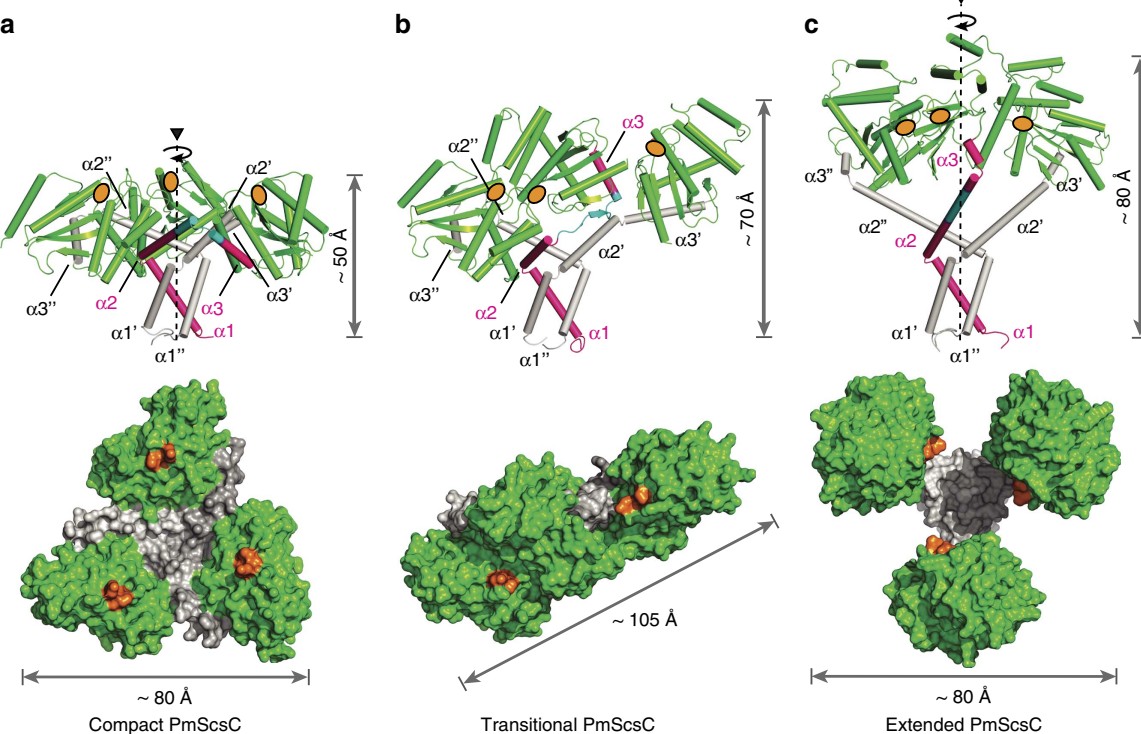

**Figure 3 | PmScsC crystal structures.** (**a**) Compact (PDB: 4XVW), (**b**) Transitional (PDB: 5IDR) and (**c**) Extended (PDB: 5ID4) crystal structures of PmScsC. Upper panels: side view, secondary structure with catalytic domains in green, 11-residue peptide linker in cyan and trimerization domains in magenta or white. Height in this orientation is indicated. In each case, one trimerization domain is shown in magenta for comparison of the conformational changes across the three crystal structures. Lower panels: top view, surface representation (catalytic TRX fold domains green, trimerization domains white), maximum dimension in this orientation is labelled. Active site positions are indicated in orange for each protomer.

| Table 1 | PmScsC crystal structure statistics. | | |
|---|---|---|---|
| | **Compact (4XVW)** | **Transitional (5IDR)** | **Extended (5ID4)** |
| *Data collection* | | | |
| Space group | P 2$_1$ | I4 | H3$_2$ |
| Cell dimensions | | | |
| $a, b, c$ (Å) | 137.5, 163.9, 181.9 | 193.1, 193.1, 105.8 | 86.7, 86.7, 330.9 |
| $\alpha, \beta, \gamma$ (deg) | 90, 90, 90 | 90, 90, 90 | 90, 90, 120 |
| Resolution (Å) | 91.15-2.60 (2.74-2.60) | 136.51-2.56 (2.57-2.56) | 110.29-2.92 (2.93-2.92) |
| $R_{merge}$ | 0.072 (0.617) | 0.083 (0.741) | 0.059 (0.625) |
| $I/\sigma I$ | 11.0 (2.0) | 14.9 (2.2) | 14.2 (2.8) |
| Completeness (%) | 98.6 (95.4) | 99.4 (100.0) | 99.2 (100.0) |
| Redundancy | 3.8 (3.7) | 4.1 (4.1) | 4.1 (4.2) |
| | | | |
| *Refinement* | | | |
| Resolution (Å) | 91.15-2.60 | 42.82-2.56 | 40.36-2.92 |
| No. reflections | 243409 | 62069 | 10652 |
| $R_{work}/R_{free}$ (%) | 24.8/28.2 | 17.1/22.2 | 25.1/26.3 |
| No. atoms | | | |
| Protein | 40850 | 10262 | 1720 |
| Ligand/ion | NA | NA | NA |
| Water | 281 | 82 | 0 |
| *B factors* (Å$^2$) | | | |
| Protein | 59.7 | 50.6 | 122.2 |
| Ligand/ion | NA | NA | NA |
| Water | 41.5 | 43.0 | NA |
| RMS deviations | | | |
| Bond length (Å) | 0.006 | 0.008 | 0.010 |
| Bond angles (deg) | 1.21 | 1.05 | 1.17 |

Single crystals were used to collect each data set. Values for the highest resolution shell are shown in parentheses.

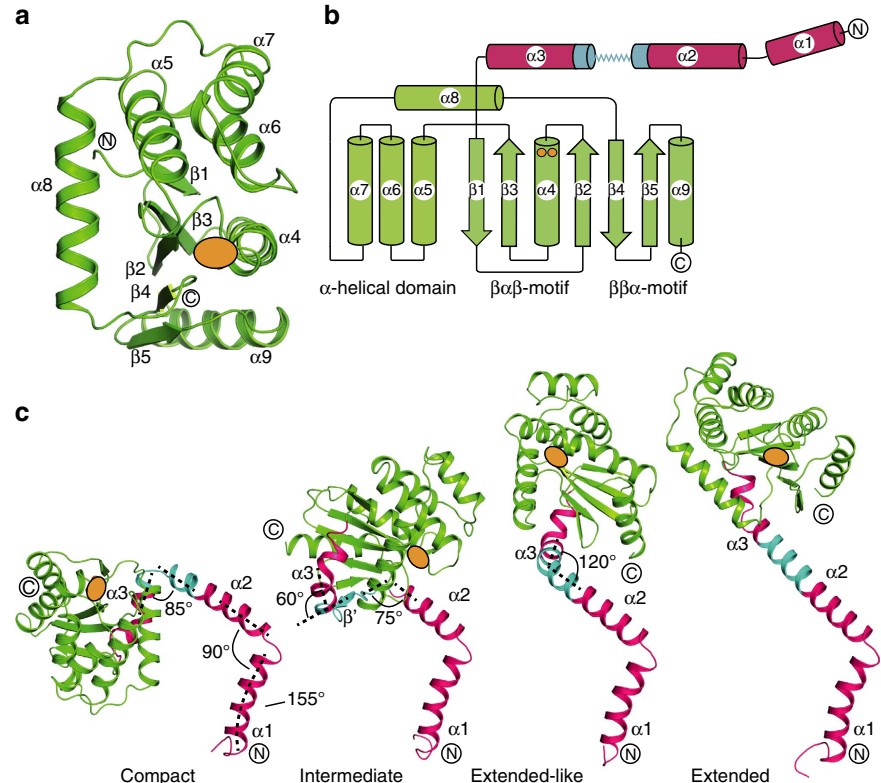

**Figure 4 | PmScsC protomer structure.** (**a**) Cartoon representation of the PmScsC catalytic domain (equivalent to a DsbA fold, green). Secondary structure elements and N- and C- termini are labelled. Position of the active site CXXC motif is shown as an orange oval shape. (**b**) PmScsC topology diagram showing the N-terminal helices α1/2/3 that precede the catalytic domain (magenta, flexible linker shown in cyan), and the DsbA domain (green) and CXXC active site (orange dots). Secondary structure elements and N- and C- termini are labelled. (**c**) The PmScsC protomers are structurally diverse: compact, intermediate, extended-like and extended. Colour scheme is as for **b**. N-terminal helices α1/2/3 and angles between the helices are labelled. N- and C- termini are also labelled.

As a consequence of the linker peptide flexibility, the three crystal structures describe extraordinarily different PmScsC quaternary shapes and dimensions. The eight compact PmScsC trimers are mushroom-shaped with dimensions $80 \times 80 \times 50 \,\text{Å}^3$ (Fig. 3a). The two intermediate PmScsC crystal structures are asymmetric flat triangles with dimensions $105 \times 40 \times 70 \,\text{Å}^3$ (Fig. 3b). The extended PmScsC crystal structure is three-leaf-clover shaped with dimensions $75 \times 80 \times 80 \,\text{Å}^3$ (Fig. 3c).

We hypothesized that the dynamic motion suggested by the crystal structures, and supported by solution data, is critical to the mechanism of PmScsC. Indeed, replacement of the shape-shifting peptide with a rigid helical peptide (PmScsC RHP, 39-AEAAAKEAAAKA-50)[12] reduced the flexibility of the protein as evidenced by small angle scattering data (Supplementary Fig. 7), and abolished activity in the isomerase assay (Fig. 1c). Furthermore, we showed that trimerization of PmScsC is critical for activity because removal of the N-terminal residues (PmScsCΔN) also abolishes *in vitro* disulfide isomerase activity (Fig. 1c) and *in vivo* complementation assays (swarming in the presence of copper) (Fig. 1b).

Curiously, Nature has also performed the ScsCΔN experiment. PmScsC shares 58% sequence identity with *Salmonella enterica* serovar Typhimurium StScsC[13] (Supplementary Fig. 8) and both proteins contribute to copper resistance. Whereas we showed above that PmScsC is required for swarming motility under copper stress, StScsC is required for growth of *S. Typhimurium* in rich media in the presence of copper. By contrast, deletion of *scsC* in *P. mirabilis* had no effect on growth in rich media whether copper was present or absent (Supplementary Fig. 2c). These findings imply that the two proteins have different target

substrates and/or different molecular functions. Indeed, StScsC and PmScsC are very different proteins. StScsC lacks the N-terminal residues of PmScsC, it is monomeric rather than trimeric, and it has no isomerase activity. Intriguingly, the catalytic domains of the two proteins (as assessed by crystal structures) are very similar (PDB: 4GXZ) (RMSD 1.0–1.6 Å for 129 Cα). On the basis of this comparison, we predicted that removal of the N-terminal trimerization residues from PmScsC would not only convert the trimer into a monomer and abolish isomerase activity, it might also convert the enzyme into a dithiol oxidase. Indeed, we confirmed that this was the case (Supplementary Fig. 1d).

These two proteins, trimeric disulfide isomerase PmScsC and monomeric dithiol oxidase StScsC, are encoded in similar loci and are both associated with copper resistance. Yet, they have very different architectures, and different molecular and cellular functions. To examine if the region encoding the N-terminal extension of PmScsC is broadly conserved in *P. mirabilis*, the *scsC* gene was PCR amplified from 25 randomly selected clinical isolates and sequenced. We found that the Pm*scsC* sequence was conserved in all 25 isolates—including the region encoding the trimerization domain and shape-shifting motif—except for a single synonymous mutation in the signal sequence. This high degree of conservation suggests a highly conserved functional role. *Caulobacter crescentus* ScsC also has an extended N-terminal region and is predicted to be dimeric[14]. Altogether with our work, this suggests that the N-terminal region of this protein is necessary for oligomerization and that there are at least two very different molecular and functional classes of ScsC.

How does the trimeric and highly dynamic molecular architecture of PmScsC compare with other structurally characterized disulfide isomerases? Until this point, highly efficient disulfide isomerases were thought to function through the presence of two TRX domains (each with a CXXC active site) that can simultaneously interact with two substrate cysteines[15]. The two active sites of V-shaped DsbC and U-shaped PDI are embedded within a dynamic framework that enable shuffling of substrate disulfides[16–19] (Fig. 5a,b). Trimeric PmScsC also has the potential to interact with multiple substrate cysteines simultaneously. However, the evidence presented here shows a much more expansive range of motion than observed previously for DsbC or PDI, as depicted in morphing movies (Supplementary Movies 2–4) and static structural comparison (Fig. 5c). For example, the distance between active sites of DsbC (PDB: 1JZD-1EEJ)[3,16] or PDI (PDB: 4EKZ-4EL1)[18] in reported crystal structures ranges between 31–39 Å (26% increase) and 28–40 Å (43%), respectively. By comparison, this distance ranges from 16 to 52 Å (225%) in PmScsC.

## Discussion

Why is it that other characterized disulfide isomerases do not have the highly flexible, trimeric structure of PmScsC? How does the monomeric *S.* Typhimurium ScsC—not a disulfide isomerase—confer copper resistance? How do the very diverse ScsC proteins (monomeric, dimeric or trimeric) form redox relay systems with a predicted partner protein ScsB[14] that appears to be highly conserved in bacterial Scs gene clusters? The likely explanation for the diversity in architecture and flexibility of ScsC proteins may relate to differences in their target specificity.

The structural diversity among ScsC homologues may simply reflect the fact that each ScsC homologue has evolved to optimize disulfide reduction and/or isomerization of a distinct protein substrate or set of substrates that are sensitive to copper-induced damage. Identifying and characterizing these specific substrates, and understanding why these are not refolded or reduced by other periplasmic TRX-fold proteins are key questions that will inform our understanding of bacterial copper resistance mechanisms.

In summary, PmScsC is a unique and highly dynamic disulfide isomerase. The N-terminal residues bring three catalytic domains together into a trimer, and a unique flexible linker enables extraordinary twisting and extending motions that impact on the catalytic domain placement. Crystal structure snapshots supported by X-ray scattering data from solution, indicate an almost doubling in length of the protein in one dimension, in concert with considerable catalytic domain rotation. The combination, extent and diversity of these motions would enable mis-folded substrates bound to PmScsC to explore a broad folding landscape, consistent with the redox foldase activity of this key *P. mirabilis* copper resistance protein. These findings show in much greater detail than ever before how redox proteins with multiple active sites are able to shuffle incorrectly folded substrates. It is unclear whether PmScsC directly interacts with copper or how it interacts with other proteins required for virulence under copper stress. Nevertheless, the structural data may provide a suitable basis for drug discovery targeting a central defence mechanism of an important human pathogen[20,21]. Importantly, the shape-shifting motif may also represent a modular component for dynamic motion that could be used in 'plug and play' protein engineering applications.

## Methods

**Bacterial strains and growth conditions.** BLAST searches of DsbA homologues were performed on the genome of *Proteus mirabilis* HI4320 (ref. 22). Clinical isolates of *P. mirabilis* were cultured from the urine or blood of patients with urinary tract infection or bacteremia. The isolates were sourced from the Princess Alexandra Hospital or Sullivan Nicolaides Pathology (Brisbane, Australia). *E. coli* Top10 (Invitrogen) was used for plasmid manipulations. *E. coli* and *P. mirabilis* were cultured in Luria-Bertani (LB) broth (10 g l$^{-1}$ tryptone, 5 g l$^{-1}$ yeast extract, 10 g l$^{-1}$ NaCl) on LB agar (LB medium containing 15 g l$^{-1}$ agar). To prevent swarming of *P. mirabilis*, LB medium containing 30 g l$^{-1}$ agar and without NaCl was used. Media were supplemented with chloramphenicol (30 μg ml$^{-1}$), ampicillin (100 μg ml$^{-1}$) or kanamycin (25 μg ml$^{-1}$) as required.

**Construction of *Proteus mirabilis scsC* mutants.** Mutation of the *scsC* gene in *P. mirabilis* PM38 and PM54 was performed using the TargeTron gene knockout system (Sigma-Aldrich) as per the manufacturer's instructions. Briefly, optimal intron insertion sites and primer sequences (5827, 5′-aaaaaagcttataattatc cttaag-tatcgaagatgtgcgcccagataggggtg, 5828, 5′-cagattgtacaaatgtggtgataacagataagtcgaa-gatgctaacttacctttctttgt, 5829, 5′-tgaacgcaagtttctaatttcgattatacttcgatagagg aaagtgtct) were predicted using the Sigma TargeTron online algorithm, followed by a retargeting PCR and cloning of the amplicon into the pACD4-K-C shuttle vector, resulting in the generation of plasmid pACD4-K-C*scsC*. Plasmid pAC4K-C*scsC* and pAR1219 (helper plasmid) were transformed into PM38 and PM54; induction and retargeting of the intron containing the kanamycin-resistance gene cassette into *scsC* was performed according to the manufacturer's instructions. Transformants with the correct intron insertion were selected by growth in the presence of kanamycin. Mutants with the correct insertion were confirmed by PCR and nucleotide sequencing using primers 5914 (5′-atgaaaaaaatttggctggcgttagc) and 5915 (5′-ttatttttttcatcagtaattgattgacgacatcagaataag), and referred to as PM38*scsC* and PM54*scsC*, respectively. The reprogrammed intron was inserted between nucleotide 528 and 529 in the sense strand of *scsC*. Plasmids pACD4-K-C*scsC* and pAR1219 were cured by passaging on nonselective medium followed by screening for loss of chloramphenicol and ampicillin resistance. Plasmid pScsC was constructed by amplifying the *scsC* gene from PM54 using primers PMscsC-pSU2718F (5′- gcgctctagattaactattctttcagaggctaaaggagcc) and PMscsCpSU2718R (5′-cgaagcttcgttattttttcactttcgccagttgttctttcac), digestion with XbaI-HindIII and ligation into XbaI-HindIII digested pSU2718. Mutation of the CXXC active site (C82A; plasmid pScsC AXXC) and deletion of residues 1–42 (plasmid pScsCΔN) was performed by PCR; all constructs were confirmed by sequencing. These mutants retained the periplasmic signal sequence. Complementation was performed by

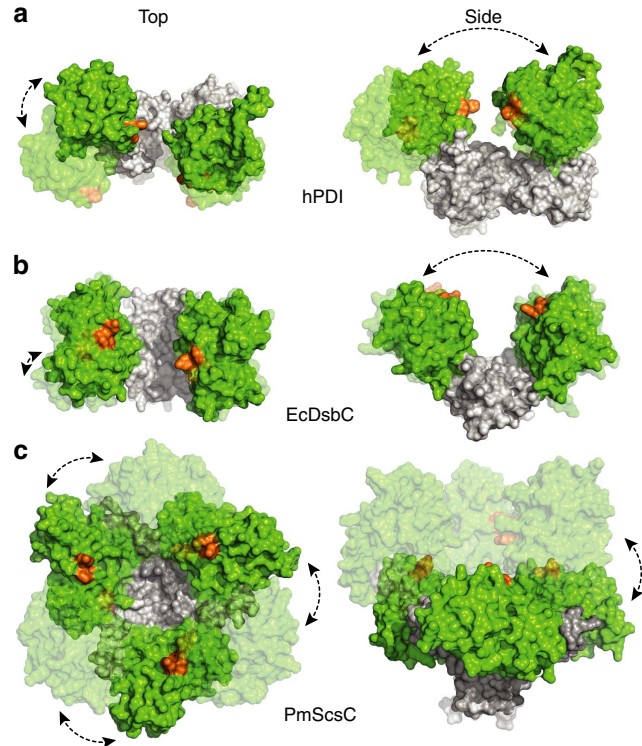

**Figure 5 | Disulfide isomerase range of motion.** Top and side views of (**a**) U-shaped human PDI with two catalytic domains (PDB: 4EKZ and 4EL1), (**b**) V-shaped homodimeric EcDsbC with two catalytic domains (PDB: 1JZD and 1EEJ) and (**c**) homotrimeric PmScsC (this work, PDB: 4XVW and 5ID4) showing its expansive twisting and elongation motions. Catalytic domains are shown in green, active sites (CXXC) in orange, and oligomerization or non-catalytic domains in white. See also Supplementary Movie 1.

introducing plasmids pScsC, pScsC AXXC, pScsCΔN or pSU2718 (vector control) into PM54*scsC*.

**Swarming motility in the presence of CuSO₄.** Overnight cultures of *P. mirabilis* were diluted in fresh LB broth to an $OD_{600}$ of 0.1. Swarming on LB agar was examined in the absence or presence of $CuSO_4$ (final concentration of 1.5 mM). For assays performed in the presence of $CuSO_4$, a 10 μl sample of the diluted *P. mirabilis* PM54 and PM54*scsC* strains, respectively, was inoculated onto the centre of an agar plate and allowed to soak into the agar (resulting in equal diameter for each sample). The plates were incubated at 37 °C. The diameter of the swarming zone was measured after 14 h incubation, and every one or two hours thereafter for a total of 10 h. Swarming of PM54*scsC* containing pSU2718 (vector control) or complementation plasmids (pScsC, pScsC AXXC and pScsCΔN) was performed as described above, but with the addition of chloramphenicol (30 μg ml⁻¹) and 1 mM isopropyl β-D-1-thiogalactopyranoside for ScsC induction. Swarming motility at 37 °C on LB agar was measured after 6 and 8 h of incubation. Three replicate experiments were performed for all assays. Swarming motility of PM38 wild-type, mutant and complemented strains in the presence and absence of copper was performed in the same manner.

**Western blotting.** ScsC expression in *P. mirabilis* wild-type, mutant and complemented strains was examined by western blot analysis. Crude cell lysates were prepared from standardized overnight cultures. Samples were resolved by SDS–polyacrylamide gel electrophoresis (SDS–PAGE) and transferred onto a polyvinylidene difluoride membrane. Blots were probed using a PmScsC specific polyclonal antiserum raised in rabbits (WEHI Antibody Facility) at 1:1,000 dilution in phosphate buffer saline with 0.05% *v/v* Tween-20.

**Protein production.** Codon-optimized PmScsC (see Supplementary Note 1 for the sequence) lacking the first 21 amino acids (which correspond to the predicted secretion signal) or PmScsCΔN (lacking the first 63 residues, starting at $A^1Q^2F^3...$) was inserted into pMCSG7 (Midwest Center for Structural Genomics) by ligation-independent cloning. PmScsC and PmScsCΔN were expressed in BL21(DE3)pLysS cells (Novagen) at 30 °C for 16–20 h using ZYP-5052 autoinduction media[23]. Selenomethionine-labelled PmScsC (mature PmScsC contains four native methionine sites) was expressed in BL21(DE3)pLysS cells grown in minimal media (M63) supplemented with 0.05 mg ml⁻¹ D/L-selenomethionine. Recombinant protein expression was induced by 0.1 mM isopropyl β-D-1-thiogalactopyranoside overnight at 30 °C.

After expression, all cells were collected by centrifugation then resuspended in lysis buffer (25 mM Tris pH 7.4, 150 mM NaCl) with added protease inhibitor (1:1,000 dilution into lysate, BioPioneer, Inc., USA) and DNase I (3.3 μg ml⁻¹ final concentration, Sigma-Aldrich). The bacterial cells were processed through a Constant Systems (LTD, UK) TS-Series cell disruptor twice, at pressures of 22 and 24 kpsi. The lysate was clarified by centrifugation (40,000 *g*, 30 min, 4 °C, rotor JA-25.5, Beckman Coulter, Brea, CA) and the soluble fraction was run twice over 2 × 3 ml of equilibrated Talon (Clontech) resin to bind the His-tagged protein. The resin was washed with six column volumes of lysis buffer before the protein was eluted with 25 mM Tris pH 7.4, 150 mM NaCl, 250 mM imidazole. To remove the His₆-tag, Tobacco Etch Virus (TEV) protease cleavage was performed at room temperature for 2 h with 1 mg of TEV protease per 50 mg of His-tagged protein. The protein mixture was then desalted using a Sephadex G-25 fine 16/60 column connected to an ÄKTA FPLC system (GE Healthcare) and reverse IMAC (2 × 2 ml of Talon resin) was used to separate cleaved protein from the His-tagged contaminants. PmScsC and PmScsCΔN were reduced or oxidized with a 25-fold molar excess of DTT, or a 10-fold molar excess of copper(II)/1,10-phenanthroline, respectively. The final step of purification for the proteins was size-exclusion chromatography using a Superdex S75 column with 10 mM HEPES pH 7.4, 150 mM NaCl. Proteins were concentrated using Amicon Ultra centrifugal filter devices with a 10-kDa cutoff (Merck Millipore, USA). Yields for PmScsC were ∼80 mg of purified protein per litre of culture. SDS–PAGE (NuPAGE 4–12% BisTris gel, Invitrogen, Australia) with Coomassie blue stain was used to assess the protein quality. Concentration of protein samples was determined using the $A_{280}$ of the sample (read using a Thermo Scientific NanoDrop 2000c spectrophotometer) and calculated extinction coefficients from ProtParam[24].

*E. coli* DsbA and *E. coli* DsbC, lacking the periplasmic leader signal were expressed and purified, as described for PmScsC.

**Design of PmScsC rigid helical peptide mutant.** The 11-amino-acid flexible linker peptide in PmScsC (residues 39-KADEQQAQFRQ-49) was replaced with a 12-amino-acid rigid helical peptide (39-AEAAAKEAAAKA-50)[12] using overlap extension PCR[25] with the primers, 5′-gcagaagcagcggccaaagaggctgcggctaaagccgca ctggctagcgaacatgatgcc and 5′- ggctttagccgcagcctctttggccgctgcttctgctttcgtctgcagagcca tgattgc. The mutated gene was then reinserted into pMCSG7 by ligation-independent cloning. The construct was confirmed by sequencing and the protein expressed and purified, as described for wild-type PmScsC.

**PmScsC sequence conservation.** All clinical isolates were de-identified, and individual informed consent was not required. Genomic DNA isolation was performed using the UltraClean Microbial DNA Isolation Kit (MO-BIO Laboratories, #12224) according to the manufacturer's instructions. Yields varied between 100 and 500 ng μl⁻¹ of DNA. PCR amplification of the *scsC* gene from *P. mirabilis* strains was performed using a sequence specific primer pair (5′-gtgccgttt aacca-gatttatg) and (5′-cgtagataaatcagtaagttctg) in combination with Phusion High-Fidelity DNA Polymerase (New England BioLabs Inc., MA, USA). DNA was initially denatured in a single step at 98 °C for 30 s followed by 30 cycles of denaturation (98 °C, 10 s), annealing (50 °C, 20 s) and elongation (72 °C, 30 s). In a final step fragment elongation was completed at 72 °C for 7 min. PCR samples were examined by electrophoresis and subsequent DNA sequencing.

**Chemical cross-linking.** In total 150 μM of purified PmScsC (in 50 mM HEPES buffer pH 8.5) was reacted with 3 mM homobifunctional dithiobis(sulfosuccinimidylpropionate) (DTSSP) (Pierce) for different time intervals (30 s–30 min) at 23 °C. DTSSP contains one amine-reactive *N*-hydroxysulfosuccinimide (sulfo-NHS) ester at each end of its eight carbon spacer arm. The sulfo-NHS group reacts preferentially with amines (from lysine residues) between pH = 7 and 9, to form a stable amide bond. DTSSP contains a disulfide bond in the spacer arm, which can be readily cleaved with dithiothreitol (DTT). Aliquots were removed at various time points and mixed with $(NH_4)HCO_3$ (50 mM final), to stop the reaction, through hydrolysis of remaining functional ester groups in DTSSP. The samples were then run on a non-reducing SDS–PAGE and subjected to Coomassie staining. As control, 50 mM DTT was added to selected samples, which led to cleavage of the internal disulfide bond in DTSSP, and consequently disruption of cross-links between ScsC protomers.

**MALLS.** A combined approach, using analytical size exclusion chromatography (SEC) and multiangle laser light scattering (MALLS) was utilized to determine and compare the stoichiometry of PmScsC with PmScsCΔN in solution. The setting consisted of an LC20 high-performance liquid chromatography (HPLC) system (Shimadzu, Rydalmere, Australia) and a DAWN HELEOS II laser light detector connected to an Optilab T-rEX refractive index detector (Wyatt Technology, Dernbach, Germany). Either a Superdex 200 or Superdex 75 10/300 GL analytical column (GE Healthcare, USA) was connected to the LC20 HPLC system and equilibrated with 25 mM Tris and 150 mM NaCl, pH 7.5 overnight. Purified proteins were injected (500 μl of 3 mg ml⁻¹ at a flow rate of 1.0 ml min⁻¹) into the SEC-MALLS system for analysis. To calibrate the detector, 500 μl of 5 mg ml⁻¹ bovine serum albumin (BSA) (Sigma-Aldrich, Australia) was used in 25 mM Tris and 150 mM NaCl, pH 7.5 at a flow rate of 1.0 ml min⁻¹. Wyatt Astra V software was used for data collection and analysis.

**SAXS.** Small angle X-ray scattering data were collected on the SAXS-WAXS beamline at the Australian Synchrotron[26]. Immediately before loading, all samples were centrifuged at 10,000 *g* to remove large particles from the solution, and radiation damage was minimized by flowing samples (∼100 μl) past the beam in 1.5 mm quartz capillaries (Hampton Research). Data reduction was carried out using the Australian Synchrotron Scatterbrain software[27] correcting for sample transmission and solvent scattering.

Scattering data were collected on reduced PmScsC wildtype and RHP mutant proteins in 25 mM HEPES pH 7.5, 150 mM NaCl, 1 mM DTT. Data were also collected on oxidized wildtype PmScsC, but as no significant differences in the scattering were observed, only data from the reduced forms are presented. To confirm that the pH and ionic strength of the crystallization conditions do not induce conformational changes in PmScsC, scattering data were collected from wildtype PmScsC prepared at 0.50 mg ml⁻¹ in a gradient of 25 mM HEPES pH (6.0, 6.5, 7.0, 7.5 and 8.0) and NaCl (150, 300, 600, 900 and 1,500 mM). No significant systematic change in the structural parameters was observed at any point of the gradient, hence, it was concluded that neither ionic strength nor pH influence the conformation of the complex in the ranges measured.

Data quality was assessed by inspection of the linearity of the Guinier region of the data ($qR_g < 1.3$), estimated molecular mass of the protein complex, and concentration dependence of the scattering. The estimated molecular mass was determined as outlined in[28], where the contrast and partial specific volume were estimated from the protein sequence[29]. The pair-distance distribution function ($p(r)$) was generated from the experimental data using *GNOM*[30] from which $I(0)$, $R_g$ and $D_{max}$ were determined. Rigid body modelling of the scattering data from PmScsC wildtype and mutant proteins was performed using *CORAL*[31]. $C_3$ symmetry was assumed, and the starting model was oriented such the 3-fold axis was parallel with the *z*-axis, and passed through the centre of the oligomization domain. Two rigid bodies were defined for each monomer: residues 3–44 (oligomerization domain); 47–224 (catalytic domain). The position of the oligomerization domain was fixed, and the position and orientation of the catalytic domain was then optimized against the measured scattering data. The program was run 16 times for each protein, but the models with the lowest penalty function still showed small systematic deviations from the experimental data (Wildtype: $\chi^2 = 3.4$; CorMap test[32], 182 points, $C = 51$, $P = 0.000$; RHP mutant: $\chi^2 = 5.6$; CorMap test, 171 points, $C = 28$, $P = 0.000$). Given the crystal structures show

significant structural diversity, it is likely that these systematic deviations arise from the fact that there is an ensemble of structures present in solution. Hence, ensemble optimization was also performed with the program EOM[33]. From an initial pool of 1,000 structures (the oligomerization domain possessed $C_3$ symmetry in all structures, but the entire trimer was permitted to adopt either $C_1$ or $C_3$ symmetry), a final ensemble of 6 structures was obtained that yielded an excellent fit to the data (Wildtype: $\chi^2 = 1.0$; CorMap test, 182 points, $C = 12$, $P = 0.021$; Mutant: $\chi^2 = 1.8$; CorMap test, 171 points, $C = 11$, $P = 0.076$). The ensemble for the wildtype protein is composed of six structures and spans a diverse range of conformations. Thus, the SAXS data supports the notion that the PmScsC trimer is highly dynamic in solution. The ensemble for the mutant protein is composed of four structures, all in extended conformations. The initial pool of random structures was the same for both WT and RHP optimizations, and the number of structures in the final ensemble was not artificially limited. The reduction in size of the ensemble (from 6 for WT to 4 for RHP) is consistent with the RHP mutation rigidifying the flexible helix, and reducing the conformational space sampled by the protein. Details of the data collection and structural parameters are summarized in Supplementary Table 1.

**Crystallization and structure determination.** Crystallization screenings were performed at the UQ ROCX facility at the University of Queensland (uqrocx.im-b.uq.edu.au) using commercial screens and the hanging-drop vapour diffusion method.

*Compact structure.* Selenomethionine-labelled PmScsC crystals were grown at 20 °C from a drop comprising 200 nl of 30 mg ml$^{-1}$ purified protein in 10 mM HEPES pH 7.4 and 200 nl of well solution, 2.85 M sodium malonate and pH 5.8 containing 0.1 M cobalt(II)chloride hexahydrate. Crystals grew over a period of 2–3 days. A data set at the Se-edge (wavelength = 0.9792 Å) was measured under cryogenic conditions (100 K) at the Australian Synchrotron MX2 beamline using the BluIce interface[34]. Data were processed in XDS[35] and scaled to space group $P2_12_12$ in AIMLESS[36]. A manually guided molecular replacement approach was performed using StScsC as the template (PDB ID: 4GXZ) in MOLREP[37]. Manual selection of solutions based on observed packing, allowed placing of six molecules in the asymmetric unit. This solution was improved in the subsequent MR runs in MOLREP[37], allowing the addition of eight molecules in the second trial and a total of nine molecules in the third run. A SAD-MR approach in PHENIX[38], by combining SAD phasing and the partial MR solution (nine molecules), and subsequent density modifications resulted in an electron density map with a skew value of 0.82 (1,826 residues (190 side chain built) in 250 fragments). Successive AutoBuild runs in PHENIX[38] built 1,557 residues (806 side chain) and dropped the R and Rfree values to 38% and 41.5%, respectively. Extensive manual building was required to complete the model. Molecular replacement using this model as template in PHASER[39] allowed placing of 12 molecules in the asymmetric unit. After several rounds of manual building in Coot[40] and PHENIX[38] refinements R/Rfree values dropped to 29.1 and 32.3%, but further refinement cycles were unsuccessful. Careful examination of Ctruncate[41] indicated the presence of nearly perfect pseudo-merohedral twinning (-h,-k,l) and led to a reconsideration of space group. Reprocessing the data set to a resolution of 2.6 Å in space group $P2_1$ yielded lower R-merge (0.072 versus 0.097 for data processed in $P2_12_12$). A new molecular replacement run using the reprocessed data in $P2_1$ space group and the PmScsC template with a resolution cutoff set to 3.2 Å finally identified 24 copies in PHASER[39]. Rigid body refinement in REFMAC[42] showed that refinement considering the twin operator (-h,-k,l) produced better R-factors (37 versus 42%, no twinning). After manual correction of two chains that were placed inaccurately during molecular replacement by PHASER[39], and several rounds of refinement in PHENIX[38] and REFMAC[42] the final R/Rfree values are 24.8/28.2% at 2.6 Å resolution. The Ramachandran favoured/outlier statistics for the structure are 95%/0.9%. A stereo image of representative electron density for this structure is show in Supplementary Fig. 9a.

*Transitional structure.* Crystals were grown at 20 °C from drops containing 1 μl of 48 mg ml$^{-1}$ oxidized PmScsC in 10 mM HEPES pH 7.4 and 1 μl of 2.85 M sodium malonate pH 5.8 containing 0.1 M copper (II) chloride. Crystals formed after 2–3 days and were cryoprotected with 3.4 M sodium malonate pH 5.8 and flash frozen in liquid nitrogen. Data were collected using the BluIce software[34] on the MX2 beamline of the Australian Synchrotron at a wavelength of 0.9792 Å under cryogenic conditions (100 K). The data were processed with XDS[35] and scaled in AIMLESS[36] using the autoPROC framework[43]. The space group was determined to be $I4$ and molecular replacement using PHASER[39], as implemented in the CCP4 suite[44], was performed with a single catalytic domain from the compact structure (residues 46–224). Five molecules were initially found in the asymmetric unit and after assessing the crystal packing one of these was deleted and a subsequent round of PHASER[39] was run with the altered solution as an additional input model. This resulted in the placement of 6 molecules in the asymmetric unit, which is consistent with the presence of two trimers. Manual building of the trimerization domain was performed in Coot and rounds of autobuild using Buccaneer[45] in CCP4 and refinement using REFMAC[42] and BUSTER[46,47] were performed to improve the structure. Several rounds of manual adjustment in Coot[40] and refinement in PHENIX[38] were performed to yield final R/Rfree values of 17.1 and 22.2%. Validation in MolProbity[48] was used throughout the refinement process, and the final Ramachandran favoured/outlier statistics for

the structure are 98%/0.5%. A stereo image of representative electron density for this structure is show in Supplementary Fig. 9b.

*Extended structure.* Crystals were grown at 20 °C from drops containing 1 μl of 20 mg ml$^{-1}$ oxidized PmScsC in 10 mM HEPES pH 7.4, 150 mM NaCl and 2.5 mM copper (II) chloride (added to protein solution immediately before crystallization) and 1 μl of 32% Jeffamine M-600 pH 7 in 0.1 M HEPES pH 8. Crystal formation occurred within minutes of set up and crystals continued to grow until day two. Crystals were cryoprotected with 32% Jeffamine M-600 pH 7, 0.1 M HEPES pH 8 and 20% ethylene glycol and frozen in liquid nitrogen. Data were collected using the BluIce software[34] on the MX2 beamline of the Australian Synchrotron at a wavelength of 0.9537 Å under cryogenic conditions (100 K). Using the autoPROC framework[43] the data were processed with XDS[35] and scaled in AIMLESS[36]. The Wilson B factor for the data was relatively high ($\sim$ double that of the data for the other two crystal structures). The space group was determined to be $H3_2$ and molecular replacement using PHASER[39], as implemented in the CCP4 suite[44], was performed with a single catalytic domain from the compact structure (residues 46–224). Only one molecule was found in the asymmetric unit (the trimer is generated by crystal symmetry). Manual building of the trimerization domain was performed in Coot with rounds of autobuild (Buccaneer[45]) and refinement (BUSTER[47]) performed to improve the structure. Several rounds of manual adjustment in Coot[40] and refinement in PHENIX[38], including the use of TLS, were performed to yield final R/Rfree values of 25.1 and 26.3%. Validation in MolProbity[48] was used throughout the refinement process and the final Ramachandran favoured/outlier statistics for the structure are 98%/0.5%. A stereo image of representative electron density for this structure is show in Supplementary Fig. 9c.

Residues in all three crystal structures were numbered based on their position after the TEV cleavage site in the construct. MUSTANG-MR[49,50] was used for structure alignment and RMSD calculations and PyMOL was used to create images of the structures and perform other measurements. The distance range between active sites in DsbC, PDI and PmScsC structures was measured using the Cβ positions of the more N-terminal of the two catalytic cysteines. Supplementary Movie 1 was created in Pymol by generating morphs between each of the PmScsC crystal structures, as well as morphs between each of the available structures for *E. coli* DsbC (PDB: 1JZD-1EEJ) and human PDI (PDB: 4EKZ-4EL1).

**Measurement of PmScsC redox potential and p$K_a$ values.** The redox potential of PmScsC was determined by incubating 2 μM PmScsC with 1 mM GSSG in combination with varying concentrations of GSH (4–15 mM) for $\sim$ 12 h at 24 °C in 100 mM sodium phosphate buffer, pH 7.0 and 1 mm EDTA. Trichloroacetic acid, 10% $v/v$, was used to precipitate the protein samples and the pellets were washed with ice-cold acetone. The samples were then treated with 2 mM 4-acetamido-4′-maleimidylstilbene-2,2′-disulfonic acid, in 50 mM Tris–HCl, pH 8.0, and 1% SDS, which blocks free thiols and increases the weight of the reduced protein by $\sim$ 1 kDa. The reduced and oxidized forms of PmScsC were separated on a 12% non-reducing SDS–PAGE (NuPAGE) and stained with Coomassie Brilliant Blue. The fraction of reduced protein, R, was determined via densitometric analysis of the stained SDS–PAGE gel using ImageJ[51], and the equilibrium constant $K_{eq}$ was calculated via $R = ([GSH]^2/[GSSG])/(K_{eq} + ([GSH]^2/[GSSG]))$[52]. The Nernst equation, $E^0 = E^0_{GSH/GSSG} - (RT/nF) \cdot \ln K_{eq}$, where $E^0_{GSH/GSSG}$ is the standard potential of $-240$ mV (ref. 53), was used to calculate the redox potential of PmScsC.

The pH-dependent absorbance of the catalytic thiolate anion of PmScsC was determined at 240 nm (ref. 54) using a CARY 50 UV/VIS spectrophotometer (Agilent Technologies, USA). The pH titration measurements of oxidized or reduced PmScsC (40 μM) were conducted at 22 °C in 2 ml reaction buffer (10 mM Tris, 10 mM sodium citrate, 10 mM K$_2$HPO$_4$, 10 mM KH$_2$PO$_4$, 200 mM KCl and 1 mM EDTA). Absorbance ($\lambda = 240$ and 280 nm) was measured between pH 6.5 and 1.5 in 0.25 increments. The p$K_a$ value was calculated from the fitted curves of three replicates using the Henderson–Hasselbalch equation (pH = p$K_a$ − log ((A240/A280)$_{red}$/(A240/A280)$_{oxid}$)). Redox potential and p$K_a$ experiments were each repeated three times.

**Relative stability of oxidized and reduced PmScsCΔN.** Unfolding of PmScsCΔN was monitored using the change in the far-ultraviolet circular dichroism (CD) signal[55]. The largest difference in molar ellipticity for oxidized and reduced enzymes was calculated from initial far-ultraviolet CD spectra (from 250 to 190 nm) recorded at 25 and 95 °C using a Jasco J-810 circular dichroism (CD) spectropolarimeter (Jasco, USA), respectively. The unfolding of oxidized and reduced protein (PmScsCΔN$_{ox}$ = 222.5 nm, PmScsCΔN$_{red}$ = 219.5 nm) was monitored at a constant heat rate of 1 °C min$^{-1}$ starting from 25 °C and increasing to 95 °C in a 1 mm quartz cuvette. Unfolding reactions were performed using 10 μM protein in 100 mM NaH$_2$PO$_4$/Na$_2$HPO$_4$, 1 mM EDTA at pH 7.0. Reduced enzyme samples contained 0.75 mM DTT. The CD data were converted to a fraction of folded protein via $\alpha_{folded}(T) = (\theta(T) - \theta_{unfolded})/(\theta_{folded} - \theta_{unfolded})$ and $\alpha_{folded}(T)$ was then fitted using a Boltzmann sigmoid function $1-1/(1 + e^{N(T_m - T)})$, where $N$ is a scaling factor that describes the nature of the transition and $T_m$ is the protein melting temperature. The thermal unfolding experiment was repeated three times for both the oxidized and reduced protein.

**Protein disulfide isomerase assay.** The bovine pancreatic ribonuclease RNase A contains four disulfide bonds that are necessary for its enzymatic function. Starting with 'scrambled' RNase A (in which the cysteines have formed non-native disulfide bonds), one can determine the ability of other enzymes to restore its function by spectroscopically monitoring the hydrolysis of 3′,-5′-cyclic monophosphate (cCMP), a synthetic substrate of the ribonuclease. Thus, *in vitro* disulfide isomerase activity of PmScsC was monitored by using scrambled RNase A as substrate[56]. Inactive scrambled RNase A was produced by treating 70 mg of native RNase A (Sigma-Aldrich) with 6 M guanidinium chloride and 150 mM DTT in 50 mM Tris pH 8, incubated at room temperature for 16 h. After adjusting the solution to pH 8, the unfolded protein was desalted and buffer exchanged into 100 mM acetic acid/NaOH pH 4. The presence of reduced cysteines was confirmed spectrophotometrically by mixing an unfolded RNase A sample with eight-fold higher molar concentration of 5,5-dithio-bis-(2-nitrobenzoic acid) (DTNB or Ellman's reagent), which reacts with thiols to absorb at 412 nm. The unfolded RNase A was then air-oxidized over 5 days in 6 M guanidinium chloride, 50 mM Tris pH 8.5 at room temperature in the dark. After concentration of the protein and readjustment of the pH to 8.5, the scrambled RNase A was buffer exchanged into 100 mM acetic acid/NaOH pH 4 and concentrated to 11.5 mg ml$^{-1}$ for use in assays. The absence of reduced cysteines was confirmed, as described above. The total reaction volume for the scrambled RNase A assay was 750 μl containing 100 mM sodium phosphate buffer pH 7.0, 1 mM EDTA, 8.2 μM DTT and 10 μM PmScsC, PmScsCΔN, PmScsC RHP, EcDsbC or EcDsbA and 40 μM of scrambled RNase A. At various time intervals, 50 μl sample was removed and mixed with 150 μl of 3 mM cytidine 3′,-5′-cyclic monophosphate (cCMP) to measure RNase A activity. Hydrolysis of the cyclic phosphate ester bond in cCMP was monitored at 296 nm using a Synergy H1 multimode plate reader (BioTek, USA). Samples containing native RNase A and scrambled RNase A in the absence of any other enzyme served as positive and negative controls, respectively. The disulfide isomerase assays were repeated twice for each protein.

**PmScsC dithiol oxidation activity.** The potential of PmScsC/PmScsCΔN to catalyse disulfide bond formation *in vitro* was assessed using a synthetic model peptide (CQQGFDGTQNSCK), as described before[57]. Dithiol oxidation was determined fluorometrically, using a Synergy H1 multimode plate reader (BioTek Instruments, USA) with the excitation wavelength set to 340 nm and emission to 615 nm. A 150 μs delay before reading and 100 μs reading time were used for time-resolved fluorescence. The reaction buffer contained 50 mM MES, 50 mM NaCl and 2 mM EDTA at pH 5.5. The reaction volume was 50 μl in each well of a white 384-well plate (Perkin Elmer OptiPlate-384, Part #: 6007290), including increasing concentrations (80–320 nM) of EcDsbA, EcDsbC, PmScsC or PmScsCΔN and 2 mM glutathione (GSSG) as oxidant. At last, 8 μM peptide substrate was added to initiate the reaction. EcDsbA served as positive control. As negative control, reactions in the absence of proteins were used. The initial linear portion of the raw data was used to calculate the rate of oxidation as fluorescence increase per minute. The dithiol oxidation activity experiments were repeated three times for each protein.

**Data availability.** The UniProt accession code for the *P. mirabilus* ScsC protein described in this paper is C2LPE2 and the accession codes for other proteins described in this study are H9L4C1 (StScsC), P0AEG4 (EcDsbA) and P0AEG6 (EcDsbC). Coordinates and structure factors for the three ScsC crystal structures were deposited in the protein data bank (PDB) with accession codes 4XVW (compact), 5IDR (transitional) and 5ID4 (extended). In addition to the PDB accession codes listed above, PDB accession code 4GXZ (StScsC) was used as a molecular replacement template and 1EEJ and 1JZD (EcDsbC), and 4EKZ and 4EL1 (hPDI) were used to generate Fig. 5. Scattering data and models have been deposited in SASBDB[58] with accession codes SASDB94 (wildtype PmScsC) and SASDBW6 (RHP mutant). All other data are available from the corresponding authors upon reasonable request.

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

## Acknowledgements

We acknowledge use of the UQ ROCX Diffraction Facility and of the MX1, MX2 and SAXS/WAXS beamlines at the Australian Synchrotron. We thank the support teams at both facilities for advice and expert assistance and we are grateful to G. King for chemical cross-linking support. This work was supported by the Australian Research Council through a Laureate Fellowship (FL0992138, J.L.M.), Future Fellowships (FT100100662 MAS; FT130100580, B.H.) and DECRA Fellowship (DE130101169, M.T.); and the National Health and Medical Research Council (Senior Research Fellowships to J.L.M., GNT455829 and to M.A.S., GNT1106930; project grant to M.A.S., GNT1106590).

## Author contributions

E.J.F. and F.K. performed molecular cloning of ScsC constructs for protein purification, produced the ScsC protein for biochemical and structural studies. E.J.F., F.K., L.P. and H.G.C. collected diffraction data and solved the structures, and together with J.L.M. refined the structures and analysed the data. E.J.F., F.K., B.H. and M.A.H. performed the *in vitro* assays for the wildtype protein. E.J.F. designed and generated the rigid helical linker mutant and performed the *in vitro* assays for this protein. A.W.L. constructed the *P. mirabilis* mutants, generated the complementation plasmids, performed the motility assays and examined ScsC expression by western blotting. M.E.S.A. tested the growth of wild-type and mutant strains in the presence and absence of copper. F.K. and M.T. evaluated the clinical *P. mirabilis* strains. A.E.W. designed the SAXS experiments and collected, analysed and modelled the SAXS data. J.L.M. and M.A.S. designed and directed the project. H.G.C., L.P., A.E.W., M.A.S. and J.L.M. jointly supervised the research. All authors contributed to writing the paper.

## Additional information

**Competing interests:** The authors declare no competing financial interests.

