## [Peer Review File · Nature Communications]

Reviewers' comments:

Reviewer #1 (Remarks to the Author):

Furlong and colleagues use X-ray crystallography to study the conformations of a thiol-active chaperone from the pathogen *Proteus mirabilis*. Using a wide-range of techniques, they propose to show that a thioredoxin-like protein ScsC with protein disulfide isomerase activity (PDI) contributes to *Proteus mirabilis* virulence by preventing copper toxicity. They show by X-ray crystallography and small-angle X-ray scattering that ScsC adopts a range of conformations which are important for its activity. The senior author on the paper, Jennifer Martin, is an expert in bacterial PDIs and published the first structure of a bacterial PDI, DsbA.

The structural biology results on ScsC are a strength of the paper. PDIs such as ScsC are notoriously difficult to crystallize as intact proteins due to the flexibility and conformational plasticity of the interdomain segments. While many PDIs contain two catalytically active thioredoxin-like domains (or more in a single polypeptide chain), ScsC is the first trimeric PDI. Crystallizing three different interdomain arrangements is a tour-de-force and powerful evidence for the requirement of flexibility for PDI function. The trimeric mushroom shape of ScsC contrasts with the V-shape of DsbC, a dimeric bacterial PDI, and highlights the importance of the association of multiple catalytic domains for PDI activity. The morphing videos are a useful and valuable addition to the paper.

The connection between the ScsC structure and virulence and copper resistance is less clear. ScsC gene was first identified in a copper resistance locus in *Salmonella* but the mechanism of resistance is not clear. Although one of the crystal forms reported here was obtained in the presence of added copper, it doesn't appear to directly bind. Quan et al, 2007, JBC 282:28823–28833 showed that the monomeric thioredoxin-like protein DsbA was sufficient to provide copper resistance. (See also Arredondo et al, 2008, JBC 283:31469–31476.) This would suggest that the trimeric association and flexibility of ScsC is not important for copper resistance.

The authors suggest in the first sentence of the abstract that ScsC counteracts the antibacterial effects of reactive oxygen species generation by copper but there is no evidence for this. Indeed, the authors show identical growth rates with and without copper for the ScsC mutant strain. There is also no evidence for a role of protein flexibility in the copper resistance. Instead of survival, the authors looked at *Proteus* swarming behavior and observed a small but significant decrease of swarming in the presence of copper in the ScsC mutant strain. While the mechanism is unclear, the previous publication (Kurth et al, 2014, JBC 89:19810–19822) by the authors showed that *Proteus* flagella require DsbA for their correct assembly. Is the defect in *Proteus* swarming in the absence of ScsC due to a defect in flagella assembly in the presence of copper?

The authors need to address these questions directly with more assays of the copper-dependent swarming defect. The authors should show that rescue by the ScsC- Δ N mutant does not increase swarming. Mutation of the ScsC catalytic cysteine motif should similarly fail to rescue. And DsbA should fail. The effect of the ScsC mutant containing the rigid helical peptide would be interesting to test. If they can't relate ScsC oligomerization or mobility to copper resistance, then they need to rewrite the title and abstract accordingly. They have produced an important study of domain mobility in a novel trimeric PDI but shouldn't misrepresent its role in metal resistance.

Other points:

The SAXS analysis is very nice but the authors need to show the ensembles that were used to fit the SAXS data. Are the individual structural models to be deposited as part of the SAXS datasets? Obviously as more structures are used in the fit, the significance of each additional one decreases. How was it decided that six structures would be used to model the wild-type and four to model the

RHP mutant?

Figure 1 would be improved if a small schematic of the domain organization of ScsC were added with the 11-residue peptide linker and N-terminal trimerization helices indicated.

Much of the Materials & Methods refers to supplemental figures and would seem to be more appropriately placed in the supplemental section. An insulin reduction assay is described in the M & M but not mentioned or shown. The paper would benefit from a longer format that would allow inclusion of more of the supplemental results and discussion.

The authors should temper their discussion of the difference between oxidase and isomerase activities. The distinction is much less clear-cut than implied. Studies of yeast PDI (Tian et al, 2006 Cell 124, 61–73) show both activities are conserved even with only one catalytic thioredoxin-like domain present. Saying that multiple thioredoxin-like domains are required for isomerase activity is a convenient narrative but not supported by all the literature.

Reviewer #2 (Remarks to the Author):

In their manuscript "Proteus mirabilis uses a shape-shifting redox foldase to combat antibacterial effects of copper", Furlong et al describe their characterization of a disulfide isomerase from a human uropathogen. My overall assessment of the article is positive but somewhat dampened by the prospects of further impact.

Beginning with the strengths: the direction of the study and the experimental results are clear. The investigators brought together a host of experiments, each of which individually produced clear results and together characterized the structure and function of the protein complex. The origin of the protein is of human health relevance; increasing impact. The fold shifting, ten amino acid, sequence may receive some interest for protein engineering and as a signature in sequence analysis. The team deserves credit for the excellence of combined experimental results.

My central negative is that despite having characterized structure and function, the mechanistic necessity of the unique aspects of this protein complex remains poorly elucidated. The protein is uniquely trimeric – why? A ten amino acid sequence changes fold – does it provide some mechanistic advantage over other proteins with similar function? Perhaps some discussion of targets may be warranted. Is the proteome of *Proteus mirabilis* cysteine rich? Is there one gene in particular that has di-sulfide problems? Perhaps the overall size and trimeric nature suggests something about a particular target given the limited size of viral genomes or linkage to Copper resistance. The authors suggest that structural characterization forms the basis for drug discovery but I did not find that credible from the description. In fact the sophisticated flexibility of the protein suggests one to avoid this protein for targeting. How would one break this particular enzyme within the context of infection given the structural information provided by the manuscript?

Other minor points of improvement:

The presentation of the experimental results in Figure 1 could be improved.

Figure 1, even after reading through the manuscript a few times I had a hard time figuring out what the difference is between samples from PM54 and PM54scsC. May be worth putting a hint in the figure legend. Panel C does not really go with A and B. I would lobby the editor on the basis that A and B are small and ask for five figures where figure 2 would be MALLS results and SAXS results together. The SAXS results were really important and in the current manuscript they are relegated to the supplement.

Conclusion: I'd recommend publication if the authors could provide some insights into what advantage the unique aspects of this protein complex provides the virus. The contrasts between

other similar function proteins does not address this in the current manuscript.

Reviewer #3 (Remarks to the Author):

The manuscript under review describes experiments aimed at detailed characterization of ScsC protein of *Proteus mirabilis*. Using different strategies the authors showed that PmScsC is an untypical thiol oxidoreductase involved in shuffling incorrect disulfide bonds. The authors put emphasis mainly on the structural analysis of PmScsC and documented that PmScsC is a trimeric protein highly dynamic in solution. They indicated the eleven residue peptide linking the trimerization and catalytic domain as protein fragment responsible for conformational flexibility of this foldase. Furthermore they showed that removing the N-terminal fragment of PmScsC abolishes its isomerase activity and convert the enzyme into thiol oxidase. It should be pointed out that the studies used a combination of biochemical, microbiological and biophysical methods to provide new insights into the functioning of PmScsC. The presented data are of great importance as they documented how diverse are Dsb systems operating in prokaryotic cells. However, I would expect short explanation about the mechanism which is responsible for regulation of the process. Why many other described thiol oxidoreductases being dimers as EcDsbC are able to work without so dynamic structure. What is unique for *P. mirabilis* Dsb system that this species needs so "sophisticated" protein.

The research is well conducted and fairly clearly presented. The selected methods are of high standard and relevant to the goal of the work. All conducted out experiments are well thought out and equipped with proper controls. Additionally some hypothesis are verified using complementary experimental strategies. Enclosed videos help to understand conformational changes of PmScsC. Overall, the manuscript is of high standard, however, I have some comments, which need to be addressed

1. The manuscript lacks information concerning process of PmScsC re-reduction (analysis of the *P. mirabilis* genome in terms of presence of genes encoding enzymes involved in electron transport from cytoplasmic thioredoxin such as DsbD, ScsB or CcdA). Furthermore I think that checking PmScsC redox state in wild type and mutated cells should provide significant information about mechanism of *P. mirabilis* Dsb system functioning and allow more exhaustive comparison with previously described ScsCs from *Caulobacter crescentus* and *Salmonella typhimurium*.

2. The authors showed that *P. mirabilis* strain lacking ScsC is defective in swimming motility. I think it would be informative to identify the target of PmScsC. Such experiment would provide knowledge concerning the Dsb system functioning and would tell us more about its role in virulence. The most of bacterial thiol oxidoreductases responsible for rearrangements of incorrectly paired cysteines are involved in copper resistance.

3. The title of the manuscript is inadequate to its contents. The manuscript pay emphasis mainly on PmScsC structure and its biochemical attributes not on mechanism of copper resistance.

4. I would suggest to remove the first paragraph of the text (line 40 -45). Even described thiol oxidoreductase containing flexible peptide potentially can be a target of antibacterial drug the development of antivirulence drugs is not a subject of this work. Furthermore the level of motility reduction displayed by mutated cells is rather moderate.

Reviewer #1

Furlong and colleagues use X-ray crystallography to study the conformations of a thiol-active chaperone from the pathogen *Proteus mirabilis*. Using a wide-range of techniques, they propose to show that a thioredoxin-like protein ScsC with protein disulfide isomerase activity (PDI) contributes to *Proteus mirabilis* virulence by preventing copper toxicity. They show by X-ray crystallography and small-angle X-ray scattering that ScsC adopts a range of conformations which are important for its activity. The senior author on the paper, Jennifer Martin, is an expert in bacterial PDIs and published the first structure of a bacterial PDI, DsbA.

The structural biology results on ScsC are a strength of the paper. PDIs such as ScsC are notoriously difficult to crystallize as intact proteins due to the flexibility and conformational plasticity of the interdomain segments. While many PDIs contain two catalytically active thioredoxin-like domains (or more in a single polypeptide chain), ScsC is the first trimeric PDI. Crystallizing three different interdomain arrangements is a tour-de-force and powerful evidence for the requirement of flexibility for PDI function. The trimeric mushroom shape of ScsC contrasts with the V-shape of DsbC, a dimeric bacterial PDI, and highlights the importance of the association of multiple catalytic domains for PDI activity. The morphing videos are a useful and valuable addition to the paper.

The connection between the ScsC structure and virulence and copper resistance is less clear. ScsC gene was first identified in a copper resistance locus in *Salmonella* but the mechanism of resistance is not clear. Although one of the crystal forms reported here was obtained in the presence of added copper, it doesn't appear to directly bind. Quan et al, 2007, JBC 282:28823–28833 showed that the monomeric thioredoxin-like protein DsbA was sufficient to provide copper resistance. (See also Arredondo et al, 2008, JBC 283:31469–31476.) This would suggest that the trimeric association and flexibility of ScsC is not important for copper resistance.

We agree that there are other thioredoxin (TRX) -fold proteins that are not trimeric or highly flexible that confer copper resistance in bacteria. The *Salmonella enterica* serovar Typhimurium ScsC, which was already discussed in our original submission, is one such protein. Like PmScsC it is a TRX fold protein, but it is monomeric rather than trimeric. Our explanation for this diversity in architecture is that it reflects differences in their target specificity. The structural diversity observed among ScsC homologues may simply reflect the fact that each homologue has evolved to optimise reduction and/or isomerization of a distinct protein substrate or set of substrates that is sensitive to copper-induced damage. We have added a new paragraph in the Discussion to address this question.

The authors suggest in the first sentence of the abstract that ScsC counteracts the antibacterial effects of reactive oxygen species generation by copper but there is no evidence for this. Indeed, the authors show identical growth rates with and without copper for the ScsC mutant strain.

We stated in the abstract that copper resistance is a key virulence trait of *P. mirabilis* and that PmScsC contributes to copper resistance. These statements still stand. However, to address this comment we have modified the abstract as requested.

There is also no evidence for a role of protein flexibility in the copper resistance. Instead of survival, the authors looked at *Proteus* swarming behavior and observed a small but significant decrease of swarming in the presence of copper in the ScsC mutant strain. While the mechanism is unclear, the previous publication (Kurth et al, 2014, JBC 89:19810–19822) by the authors showed that *Proteus* flagella require

DsbA for their correct assembly. Is the defect in Proteus swarming in the absence of ScsC due to a defect in flagella assembly in the presence of copper?

In the submitted manuscript, we had shown that PmScsC contributes to a key *P. mirabilis* virulence trait, the ability of the organism to swarm in the presence of copper. We now provide additional mechanistic data (see below) showing that this function is dependent on redox activity (active site cysteine), and on trimerisation with flexibility (the presence of the N-terminal 42 residues).

In Kurth *et al* (2014) we reported that PmDsbA (a monomeric protein thiol oxidase from *P. mirabilis*) complemented an *E. coli* DsbA null mutant because it supports the production of functional flagella. Complementation of *E. coli* DsbA is a common assay for confirming *in vivo* DsbA activity (protein thiol oxidase/disulfide formation). This is because *E. coli* FlgI requires a single disulfide bond for activity. Without DsbA, FlgI does not form its disulfide bond and does not fold correctly. The absence of functional FlgI results in the absence of *E. coli* flagella. While we cannot rule out that PmScsC contributes to flagella production in *P. mirabilis*, we know that PmScsC is not a thiol oxidase so if it does support *P. mirabilis* flagella production, this would likely be through an entirely different mechanism.

The authors need to address these questions directly with more assays of the copper-dependent swarming defect. The authors should show that rescue by the ScsC-deltaN mutant does not increase swarming. Mutation of the ScsC catalytic cysteine motif should similarly fail to rescue. And DsbA should fail. The effect of the ScsC mutant containing the rigid helical peptide would be interesting to test. If they can't relate ScsC oligomerization or mobility to copper resistance, then they need to rewrite the title and abstract accordingly. They have produced an important study of domain mobility in a novel trimeric PDI but shouldn't misrepresent its role in metal resistance.

In response to this reviewer's comment, we have now generated the deltaN mutant and the catalytic cysteine mutant, as requested by the reviewer. We have now shown that complementation of the null mutant by wild-type PmScsC rescues swarming activity in the presence of copper, and that both mutated versions of PmScsC fail to rescue this defect. These data, that support our contention that the trimeric PmScsC architecture, motion and redox activity all contribute to a copper resistant swarming trait, are now included in Figure 1a. We were unable to generate the rigid helical mutant in the time available, or to generate the DsbA complementation. These would be interesting experiments for the future.

Other points:

The SAXS analysis is very nice but the authors need to show the ensembles that were used to fit the SAXS data. Are the individual structural models to be deposited as part of the SAXS datasets? Obviously as more structures are used in the fit, the significance of each additional one decreases. How was it decided that six structures would be used to model the wild-type and four to model the RHP mutant?

Ensembles generated using BIOSAXS are indicative, not absolute, as described in the online user manual (<https://www.embl-hamburg.de/biosaxs/manuals/eom.html>) – “Please note that the PDB files of the selected models **are not "the structure of the flexible system"** but only models that suggest the behaviour of the system in solution”. We did deposit these ensembles, and gave the codes at the end of the SAXS methods section. We had expected that reviewers would have had access to these during the review process. However, it appears that a special link is now needed to view unreleased entries. Those codes are:

WT: <https://www.sasbdb.org/data/SASDB94/qytf9v3rds/>

RHP: <https://www.sasbdb.org/data/SASDBW6/r72p7rmkau/>

Regarding the number of structures in each ensemble, the underlying factor is flexibility. This is highlighted by a quote from the paper describing EOM, the method we used (JACS, 2007, 129, 5656-5664), “*The number of conformers in the subset reflects the flexibility of the system...*”. The initial pool of random structures was the same for both optimisations, and we did not artificially limit the number of structures in the final ensemble. Therefore, the number of structures in the final ensemble was determined entirely by the optimisation algorithm. We have modified the text in the methods section to highlight this point better, as the outcomes of the modelling support the notion that the RHP mutant is less flexible than the wild-type protein.

Figure 1 would be improved if a small schematic of the domain organization of ScsC were added with the 11-residue peptide linker and N-terminal trimerization helices indicated.

We have added a new panel to Figure 1, as requested, to show domain organisation of wild-type and mutant ScsCs to help clarify regions of the protein that were modified.

Much of the Materials & Methods refers to supplemental figures and would seem to be more appropriately placed in the supplemental section. An insulin reduction assay is described in the M & M but not mentioned or shown. The paper would benefit from a longer format that would allow inclusion of more of the supplemental results and discussion.

The reviewer indicates that the materials and methods in the main text might be more appropriately moved to supplemental material, and also suggests that the paper would benefit from including more of the supplemental results and discussion in the main text. We have modified the text and supplemental material so that the most relevant information is included in the main text.

The authors should temper their discussion of the difference between oxidase and isomerase activities. The distinction is much less clear-cut than implied. Studies of yeast PDI (Tian et al, 2006 Cell 124, 61–73) show both activities are conserved even with only one catalytic thioredoxin-like domain present. Saying that multiple thioredoxin-like domains are required for isomerase activity is a convenient narrative but not supported by all the literature.

We agree to some extent. However, we note that *in vitro* isomerase activity may not reflect *in vivo* activity (ie the presence of a redox buffer can enable oxidation/reduction cycling of substrate that can support *in vitro* “isomerase” activity by a non-isomerase redox protein such as DsbA for example). Our results and those of others support the notion that dimerization or presence of more than one catalytic domain makes *in vitro* isomerization more efficient, and is a common and important feature of disulfide isomerases in nature. However, to address this reviewers comment, we have modified the text on p8 to refer to highly efficient disulfide isomerases.

Reviewer #2

In their manuscript “Proteus mirabilis uses a shape-shifting redox foldase to combat antibacterial effects of copper”, Furlong et al describe their characterization of a disulfide isomerase from a human uropathogen. My overall assessment of the article is positive but somewhat dampened by the prospects of further impact.

Beginning with the strengths: the direction of the study and the experimental

results are clear. The investigators brought together a host of experiments, each of which individually produced clear results and together characterized the structure and function of the protein complex. The origin of the protein is of human health relevance; increasing impact. The fold shifting, ten amino acid, sequence may receive some interest for protein engineering and as a signature in sequence analysis. The team deserves credit for the excellence of combined experimental results.

My central negative is that despite having characterized structure and function, the mechanistic necessity of the unique aspects of this protein complex remains poorly elucidated. The protein is uniquely trimeric – why? A ten amino acid sequence changes fold – does it provide some mechanistic advantage over other proteins with similar function?

We have included new data confirming the mechanistic necessity of the unique aspects of this protein (see Figure 1b). As outlined in a new paragraph in the Discussion, the unique aspects of this protein are most likely a consequence of its specific or unique substrates. Identifying and characterizing these substrates is a critical question for follow-up, but will take several years and is thus beyond the scope of the present work.

Perhaps some discussion of targets may be warranted. Is the proteome of *Proteus mirabilis* cysteine rich? Is there one gene in particular that has di-sulfide problems? Perhaps the overall size and trimeric nature suggests something about a particular target given the limited size of viral genomes or linkage to Copper resistance.

The proteome of *P. mirabilis* is not extraordinarily cysteine rich, and at present we have not been able to identify specific targets of PmScsC. We agree that the overall size and trimeric nature may suggest something about the particular target substrate, and have added text to that effect into the Discussion. However, identifying and confirming the target substrate of PmScsC is not trivial and we believe a detailed discussion of possible targets in the absence of experimental data would be far too speculative. We note that *P. mirabilis* is a bacterium not a virus.

The authors suggest that structural characterization forms the basis for drug discovery but I did not find that credible from the description. In fact the sophisticated flexibility of the protein suggests one to avoid this protein for targeting. How would one break this particular enzyme within the context of infection given the structural information provided by the manuscript?

The PmScsC activity depends not only on dynamic motion, but also on the catalytic domain active site and we have already developed inhibitors targeting active sites in *E. coli* DsbA (Adams 2015 *Angew Chemie*¹; Duprez 2015 *J Med Chem*²). In response to this reviewer's comments, and that of reviewer 3, we have removed the first paragraph of the text relating to drug discovery. We have toned down the text relating to drug discovery in the Discussion, and added the two references to our drug discovery outcomes to state explicitly that inhibitors of TRX-fold catalytic domains can be developed.

Other minor points of improvement:

The presentation of the experimental results in Figure 1 could be improved. Figure 1, even after reading through the manuscript a few times I had a hard time figuring out what the difference is between samples from PM54 and PM54scsC. May be worth putting a hint in the figure legend.

We apologise for this confusion. We have modified the Figure 1 legend to clarify these points, and included a schematic as suggested by reviewer 1, to highlight more clearly the different mutants.

Panel C does not really go with A and B. I would lobby the editor on the basis that A and B are small and ask for five figures where figure 2 would be MALLS results and SAXS results together. The SAXS results were really important and in the current manuscript they are relegated to the supplement.

As suggested by the reviewer, the SAXS results showing the scattering of the native ScsC have been combined with MALLS results and included as a new figure (Figure 2).

Conclusion: I'd recommend publication if the authors could provide some insights into what advantage the unique aspects of this protein complex provides the virus. The contrasts between other similar function proteins does not address this in the current manuscript.

We note that *P. mirabilis* is a bacterium not a virus. Most likely, the unique features of PmScsC have evolved along with its specific substrate(s). As yet, we do not know what this substrate(s) is/are. Identifying substrates and characterizing their interaction with PmScsC and why the unique features of PmScsC are needed would require several more years' work. These questions are therefore beyond the scope of the present work - which has already taken up many years effort of several PhD students and postdoctoral researchers.

Reviewer #3 (Remarks to the Author):

The manuscript under review describes experiments aimed at detailed characterization of ScsC protein of *Proteus mirabilis*. Using different strategies the authors showed that PmScsC is an untypical thiol oxidoreductase involved in shuffling incorrect disulfide bonds. The authors put emphasis mainly on the structural analysis of PmScsC and documented that PmScsC is a trimeric protein highly dynamic in solution. They indicated the eleven residue peptide linking the trimerization and catalytic domain as protein fragment responsible for conformational flexibility of this foldase. Furthermore they showed that removing the N-terminal fragment of PmScsC abolishes its isomerase activity and convert the enzyme into thiol oxidase. It should be pointed out that the studies used a combination of biochemical, microbiological and biophysical methods to provide new insights into the functioning of PmScsC. The presented data are of great importance as they documented how diverse are Dsb systems operating in prokaryotic cells. However, I would expect short explanation about the mechanism which is responsible for regulation of the process. Why many other described thiol oxidoreductases being dimers as EcDsbC are able to work without so dynamic structure. What is unique for *P. mirabilis* Dsb system that this species needs so "sophisticated" protein.

These are excellent and obvious follow-up questions. The most likely explanation for these unique features is that the trimeric nature and highly dynamic motion of PmScsC is necessary for function – ie the substrates of PmScsC may require the specific PmScsC architecture and flexibility for interaction and disulfide shuffling. But what is/are these substrates? A redox partner? A periplasmic copper-binding protein or copper transporter protein? A mis-folded protein that needs disulfides shuffled? Why are they not refolded by PmDsbC? These are important questions but are beyond the scope of the present work. We have now included text around these critical follow-up questions in the discussion.

The research is well conducted and fairly clearly presented. The selected methods are of high standard and relevant to the goal of the work. All conducted out experiments are well thought out and equipped with proper controls. Additionally

some hypothesis are verified using complementary experimental strategies. Enclosed videos help to understand conformational changes of PmScsC.

Overall, the manuscript is of high standard, however, I have some comments, which need to be addressed

1. The manuscript lacks information concerning process of PmScsC re-reduction (analysis of the *P. mirabilis* genome in terms of presence of genes encoding enzymes involved in electron transport from cytoplasmic thioredoxin such as DsbD, ScsB or CcdA).

The *P. mirabilis* ScsC gene is located within a cluster of four Scs genes. This cluster also encodes ScsB, which is a predicted partner of ScsC to maintain it in the reduced form. Cho *et al.*, 2012 mBio³ showed that in *Caulobacter crescentus* ScsB keeps ScsC in the reduced form and we believe this may also be the case for *P. mirabilis*. This information is now included in the Discussion.

Furthermore I think that checking PmScsC redox state in wild type and mutated cells should provide significant information about mechanism of *P. mirabilis* Dsb system functioning and allow more exhaustive comparison with previously described ScsCs from *Caulobacter crescentus* and *Salmonella typhimurium*.

We agree that these experiments would be interesting, but disagree that they would provide the most significant information on mechanism. Instead we focused our efforts and resources over the past few months on the more critical complementation studies which provide important information on molecular mechanism.

2. The authors showed that *P. mirabilis* strain lacking ScsC is defective in swimming motility. I think it would be informative to identify the target of PmScsC. Such experiment would provide knowledge concerning the Dsb system functioning and would tell us more about its role in virulence. The most of bacterial thiol oxidoreductases responsible for rearrangements of incorrectly paired cysteines are involved in cooper resistance.

We agree this would be a very interesting question to address and have added text in the discussion to this effect. However this additional program of research is beyond the scope of the present work.

3. The title of the manuscript is inadequate to its contents. The manuscript pay emphasis mainly on PmScsC structure and its biochemical attributes not on mechanism of cooper resistance.

We have modified the title of the manuscript to address the reviewer's comments.

4. I would suggest to remove the first paragraph of the text (line 40 -45). Even described thiol oxidoreductase containing flexible peptide potentially can be a target of antibacterial drug the development of antivirulence drugs is not a subject of this work. Furthermore the level of motility reduction displayed by mutated cells is rather moderate.

As suggested by this reviewer, and in response to reviewer 2's comments, we have removed the first paragraph of the text.

References

1. Adams, L.A. *et al.* Application of fragment-based screening to the design of inhibitors of Escherichia coli DsbA. *Angew. Chem. Int. Ed. Engl.* **54**, 2179-2184 (2015).
2. Duprez, W. *et al.* Peptide inhibitors of the Escherichia coli DsbA oxidative machinery essential for bacterial virulence. *J. Med. Chem.* **58**, 577-587 (2015).

3. Cho, S.H. *et al.* A new family of membrane electron transporters and its substrates, including a new cell envelope peroxiredoxin, reveal a broadened reductive capacity of the oxidative bacterial cell envelope. *mBio* **3**, 1-11 (2012).

REVIEWERS' COMMENTS:

Reviewer #1 (Remarks to the Author):

The changes to the manuscript and explanations in the rebuttal letter have addressed my concerns.

Reviewer #3 (Remarks to the Author):

I have reviewed and accepted the detailed explanations provided by the authors .